# The geometric evolution of aortic dissections: Predicting surgical success using fluctuations in integrated Gaussian curvature

**Kameel Khabaz[1], Karen Yuan[1], Joseph Pugar[1,2], David Jiang[1], Seth Sankary[1], Sanjeev Dhara[1], Junsung Kim[1], Janet Kang[1], Nhung Nguyen[1], Kathleen Cao[1], Newell Washburn[3], Nicole Bohr[1], Cheong Jun Lee[4], Gordon Kindlmann[5], Ross Milner[1], Luka Pocivavsek[1] \***

1 Department of Surgery, The University of Chicago, Chicago, Illinois, United States of America,
2 Departments of Material Science and Engineering, Biomedical Engineering, and Chemistry, Carnegie Mellon University, Pittsburgh, Pennsylvania, United States of America, 3 Department of Biomedical Engineering, Carnegie Mellon University, Pittsburgh, Pennsylvania, United States of America, 4 Department of Surgery, NorthShore University Health System, Evanston, Illinois, United States of America, 5 Department of Computer Science, The University of Chicago, Chicago, Illinois, United States of America

\* lpocivavsek@bsd.uchicago.edu

**Data Availability Statement:** All raw segmentation masks are available on the public Github page: https://github.com/SurgBioMech/khabaz_2024/tree/main. If access to the raw DICOM level CT

## Abstract

Clinical imaging modalities are a mainstay of modern disease management, but the full utilization of imaging-based data remains elusive. Aortic disease is defined by anatomic scalars quantifying aortic size, even though aortic disease progression initiates complex shape changes. We present an imaging-based geometric descriptor, inspired by fundamental ideas from topology and soft-matter physics that captures dynamic shape evolution. The aorta is reduced to a two-dimensional mathematical surface in space whose geometry is fully characterized by the local principal curvatures. Disease causes deviation from the smooth bent cylindrical shape of normal aortas, leading to a family of highly heterogeneous surfaces of varying shapes and sizes. To deconvolute changes in shape from size, the shape is characterized using integrated Gaussian curvature or total curvature. The fluctuation in total curvature ($\delta K$) across aortic surfaces captures heterogeneous morphologic evolution by characterizing local shape changes. We discover that aortic morphology evolves with a power-law defined behavior with rapidly increasing $\delta K$ forming the hallmark of aortic disease. Divergent $\delta K$ is seen for highly diseased aortas indicative of impending topologic catastrophe or aortic rupture. We also show that aortic size (surface area or enclosed aortic volume) scales as a generalized cylinder for all shapes. Classification accuracy for predicting aortic disease state (normal, diseased with successful surgery, and diseased with failed surgical outcomes) is 92.8±1.7%. The analysis of $\delta K$ can be applied on any three-dimensional geometric structure and thus may be extended to other clinical problems of characterizing disease through captured anatomic changes.

scans is requested, then per HIPPA guidelines since this data is considered protected under US law, we will initiate an institutional data sharing agreement and once established the data can be transferred under the established federal guidelines of protected data. To initate this request please email the Human Imaging Research Office at The University of Chicago (hirohelp@bsd.uchicago.edu).

**Funding:** We acknowledge the support of the National Institutes of Health, NHLBI, R01HL159205 to LP. The Center for Research Informatics is funded by the Biological Science Division at The University of Chicago with additional funding provided by the Institute for Translational Medicine, CTSA grant number ULITR000430 from NIH. The funders had no role in study design, data collection and analysis, decision to publish, or preparation of the manuscript.

**Competing interests:** The authors have declared that no competing interests exist.

## Author summary

For decades, aortic dissections have proven among the most difficult aortic pathologies to classify. The aorta is the largest blood vessel in the human body. An aortic dissection is the appearance of an internal blister within the aortic wall. The predominant method of diagnosing an aortic dissection is with cross-sectional x-ray imaging like computed tomography or CT scans. The morphologic evolution of aortic dissections has been difficult to quantify. The pressing clinical need to better define the morphology both in terms of size and shape of aortic dissections using CT derived imaging data is based on the high rate of failure in current surgical methods of dissection repair. Current methods are largely based on a dimensional reduction of the aortic geometry from its native two-dimensional surface in three-dimensional space to a one-dimensional space-curve. We develop a robust method using differential geometry to define each aorta using its full surface geometry. Each aorta is now uniquely represented as a point in a two-dimensional shape-size feature space. This space can be used to in general follow aortic morphology from normal development (growth) to severe pathology. Moreover, we successfully use it to identify patients who had failed aortic surgeries.

## Introduction

Imaging modalities such as computed tomography (CT) provide sophisticated three-dimensional representations of the human body and are ubiquitously relied upon in clinical practice [1–4]. At its core, imaging captures anatomic changes that are linked to underlying pathologic processes. Anatomy is the spatial organization of tissues in space. As such, geometry is the natural mathematical framework to quantitate anatomy. Geometric approaches have widely been used to characterize and classify multiple diseases including pulmonary nodules [1], liver cirrhosis [2], and thyroid masses [3, 4]. In all of these examples, the physician's human eye defines the hallmark of disease by the appearance of nodularity or spiculation. Even the process of aging or healing often leads to the appearance of undulations or wrinkles, which is a departure from the baseline smooth geometry of healthy skin [5]. These examples highlight the ubiquitous classification of anatomic objects based upon the surfaces that define their boundaries. These surfaces are easily appreciated when the anatomy is directly visualized with gross inspection (such as skin or with dissection in the operating room). They also appear as level sets in cross-sectional imaging, most often x-ray based tomography, because of the intrinsic difference in x-ray absorption and scattering of various tissues. The problem of disease progression mathematically becomes the transformation of a smooth surface into a rough surface, with the appearance of multiple new length scales.

Quantifying what the human eye so easily discerns has proven challenging. Recent work has sought to translate successes from deep learning (DL) to utilize imaging for disease diagnosis and clinical planning [6, 7]. Challenges in the availability of high-quality data [8, 9], the lack of reproducibility and generalizability [9, 10], the disconnect between technical accuracy and clinical efficacy [11, 12], and interpretability of inherently "black-box" DL models have made the application of DL in medicine particularly fraught [13, 14]. Sophisticated approaches harnessing the machinery of Riemannian geometry have been successful in certain neuroanatomy problems [15]; however, they are limited to tissues with well-defined internal landmarks allowing for global coordinate systems, e.g. Talairach space or MNI coordinate system in brain imaging [16], which do not exist for cardiovascular tissues.

We take a fundamentally mathematical approach to developing imaging-based geometric descriptors of disease evolution in clinically meaningful systems. We focus on the challenging problem of classification in aortic dissections. The aorta is the largest blood vessel in the human body. It functions as a conduit carrying blood from the left ventricle of the heart to provide blood flow to the head, arms, abdominal organs, and legs [17, 18]. Mechanically, the aorta is akin to a pressurized distensible cylindrical shell; aortic pathologies are mechanical in nature, with aortic rupture leading to near-certain death. Aortic dissections are partial tears in the aortic wall. Mechanically, dissections are cracks that do not penetrate through the entire aortic thickness; rather, they propagate between the different concentric layers making up the aortic wall generating aortic blisters (medically termed the false lumen). Dr. Michael DeBakey, the pioneer of aortic dissection surgery, famously described diseased thoracic aortas as 'torturous', 'globular', 'U-shaped', 'dilated', and 'idiosyncratically varying' [19]. These qualitative descriptors underpin the complex changes in shape that accompany aortic disease and its morphologic evolution. The current standard of care for type B aortic dissection (TBAD) is anchored on the concept of preventing "dangerous" shape changes, such as aneurysmal degeneration, which contribute to worse outcomes [17, 18]. Thus, the concept of shape and pathology are well-accepted clinically; however, accurate quantification of these changes is limited. Thoracic aortic stabilization by placement of a fabric-covered stent into the diseased thoracic aorta, clinically termed Thoracic Endovascular Aortic Repair (TEVAR), is the preferred modern surgical approach to TBAD [17, 18].

Since the 1950's and DeBakey's initial work, the ubiquitous anatomic classifier clinically used to trigger aortic surgery has been maximum aortic diameter ($2R_m$, where $R_m$ is the maximum radius) [17, 18], yet clinicians have long appreciated that size by itself is an inadequate descriptor of aortic anatomy [17, 20–22] and have resorted to qualitative descriptors of aortic morphologic evolution. The urgent need for a richer description of aortic anatomy has been highlighted by several clinical trials using TEVAR, which showed that using size alone to trigger intervention did not improve mortality [23–26]. The cardiovascular surgical community realizes that better identification of patients who would benefit from TEVAR is needed to shift the post-intervention mortality curve in the positive direction [25–28]. Identifying anatomic parameters that impact the ability of an aorta to remodel [29] is instrumental in guiding therapy.

Geometry and mechanics have driven the use of maximum diameter as a singular scalar descriptor of aortic stability and, therefore, the dominant trigger of intervention. Geometrically, since aortas are generalized bent cylinders, the cylinder's radius naturally appears as a principal length scale describing the geometry. Since diseased aortas expand non-uniformly in the radial direction as a function of axial distance (or distance along the aortic centerline), the gradient of aortic radius along the centerline is maximized near the largest diameter. Clinically, largely for the ease of application, the gradient has been replaced by simply measuring the maximum diameter and assuming a uniform 'normal' aortic radius. This assumption is fraught with problems because normal aortic diameters, while being uniform for any given individual, show a broad distribution in the general population [30]. The problem is further complicated by the fact that uniform aortic growth ($\sim 0.5$ mm/year) occurs with aging [31]. Lacking appropriate normalization, the maximum diameter is only a descriptor of geometry in comparison to population means.

Standard biomechanics literature asserts that stability of dilated aortas is dominated by the Law of Laplace, where the stress ($\sigma$) is a simple linear function of internal pressure ($P$), aortic thickness ($h$), and maximum diameter ($2R_m$): $\sigma = P \cdot R_m/h$. By Laplace, larger aortas will have linearly proportional larger stresses. Since aortic mechanical stability is a fracture problem, once the wall stress supersedes a critical stress, the aorta will crack. Given the distribution of

$R_m$ values in the population, $\langle R_m \rangle + \delta R_m \sim 1.25 \pm 0.5$cm, and the fact that $R_m$ is not constant with time, $\partial_t \langle R_m \rangle = 0.5$mm/year, the calculation of a broadly applicable critical stress becomes nearly impossible. Furthermore, the mathematical derivation of Laplace's law assumes homogeneous isotropic shells of uniform thickness, which is not true even for non-diseased aortas [32]. More advanced mechanical failure analysis of aortic aneurysms and dissections utilizes finite element approaches (FEA); however, these ultimately lack the simple relationships between geometry (derived from clinical imaging data) and stress contained in Laplace's law. As such, their wide clinical application has been limited [33, 34]. Nearly all clinical data gathered about aortas is imaging-based geometric information. The herculean efforts to translate intricate mechanical stability models using FEA to clinical practice largely failed due to lack of interpretability and understanding of these models by practicing physicians. This failure should be heeded when developing sophisticated AI-based deep-learning models, which take raw CT scan data as inputs; the challenge of interpreting such 'black-box' models in medical applications is well known [13, 14].

We derive and validate a mathematical descriptor of aortic shape; a single variable that scales appropriately with DeBakey's famous qualitative descriptors of aortic shapes [19, 20]. As D'Arcy Thompson wrote a century ago: "In a very large part of morphology, our essential task lies in the comparison of related forms rather than in the precise definition of each" [35]. The problem of morphogenesis is deeply linked to the stability of self-reproducing structures; Rene Thom postulated that any morphological process is divided into islands of determinism, dominated by morphogenic fields whose precise mathematical form is defined by topologic homeomorphisms, separated by regions of instability and indeterminacy [36]. Topology is now a fundamental tool in the classification of rapidly changing shapes and structures in developmental biology, where anatomic descriptions are translated into topological language allowing the quantification of purely geometric transformations in growing tissues [37–39]. The generalization of a tissue or body into sets of smooth, closed, orientable surfaces (i.e. membrane systems) makes them topological objects [36, 39]. Local topological surgeries, cutting or gluing, lead to global topological changes in biological forms through regions of topological catastrophes [36, 38, 39]. We apply such methods to the study and generalization of aortic morphogenesis throughout the life-cycle of an aorta (see Fig 1).

We work with aortic surfaces derived from cross-sectional imaging such as CT scans from both normal patients and patients with aortic dissections at various stages of disease. Fig 1 outlines the problem of defining a continuous geometric parameter space that can smoothly evolve from a normal pediatric aorta to the most diseased aortic dissection. Visual inspection of the objects in Fig 1 easily leads to a qualitative description of their anatomy: the aortas grow left to right with an initial rapid change followed by a more subtle increase in size (scale), a trend that reverses when surface shape is considered. The rapidly growing aortas are self-similar concerning their shape (bent cylinders with the toroidal aortic arch followed by the cylindrical descending thoracic aorta); a shape symmetry-breaking transition occurs as the aorta further moves towards the right. We follow this transition by projecting the CT-derived anatomy of 302 aortas, distributed from normal to diseased states, into a novel geometric space spanned by orthogonal axes for shape and size defined using the shape operator and total curvature of the aortic surface. The local geometry of the aortic surface is defined by its principal curvatures, $k_1$ and $k_2$. Specific measures of aortic shape in the literature have focused on functions of mean ($\kappa_m = (k_1 + k_2)/2$) or Gaussian ($\kappa_g = k_1 k_2$) curvature [40, 41], cross-sectional deviations [42], and calculations on centerline reductions of the aortic surface [21, 43, 44]. A mathematically robust and reproducible definition of aortic shape does not exist despite these efforts because of spatial over-reduction and convolution of shape and size information. Centerline calculations lose information when the aortic surface deforms heterogeneously [45],

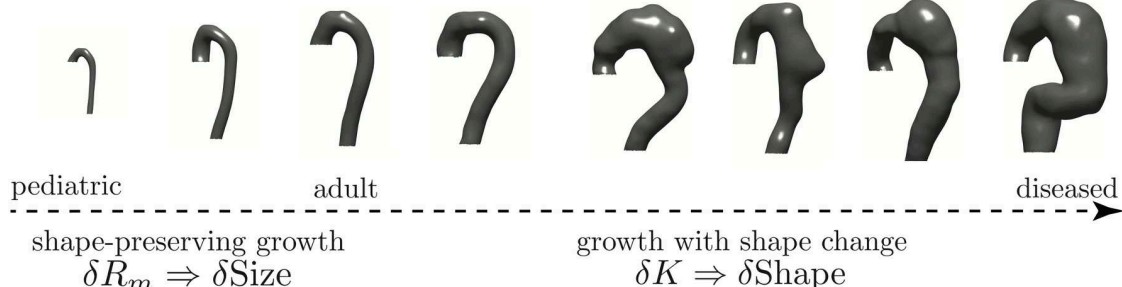

**Fig 1. Morphologic evolution of aortic shapes.** Eight representative aortas along the normal-to-diseased axis (left to right): a 3-year-old child, healthy adults, and type B aortic dissection (TBAD) patients at varying degrees of aneurysmal degeneration. Two clinical regimes exist: shape preserving growth and growth with shape changes.

and $\kappa_g$, the dominant shape function, proves itself an ill-defined shape measure because it convolutes shape and size changes [45]. We solve this problem by using the Gauss-Bonnet Theorem and the total curvature, $K = \iint_A \kappa_g dA$, as the primary measure of shape. We prove that all aortic shapes are homeomorphic to $T^2$; as such, all aortas, no matter how deformed, remain generalized bent cylinders. Using simulated shapes with evolving surface roughness, we show that $\kappa_g$ and $K$ hold the same shape information, provided overall changes in size are small.

Topology preservation is by far not the norm in biologically growing surfaces that are non-conservative and can locally add mass or remove mass without the constraint of elastic deformations or linkage to some global manifold. The existence of a homeomorphism implies that as aortas deform through normal growth or pathology, every increase in $K$ in some region must be balanced by a proportional decrease somewhere else on the aortic surface. Therefore, the distribution of $K$ across the aortic surface holds information about shape. This allows us to classify shape by studying the statistical properties of these distributions. Since $\langle K \rangle$ is constant, the variance of $K$ captures the balance of positive and negative curvature regions across the surface: $\delta K = \langle K^2 \rangle - \langle K \rangle^2$. $\delta K$ captures the heterogeneous morphologic evolution by characterizing local shape changes. We hypothesize and demonstrate that this novel shape measure correlates with aortic disease evolution and can be used to characterize treatment response for aortic dissections when appropriately coupled with a size metric.

## Methods

### Clinical data cohort

We analyzed a cohort of 302 computed tomography angiography (CTA) scans from 2009 to 2020. The non-pathologic cohort included 171 scans from 93 patients. The diseased cohort included 131 scans (SI Demographic Information), representative of two pathologies: TBAD (124 scans) and thoracic aortic aneurysms (7 scans). Three patient subgroups are derived from the main cohort to analyze geometric predictors of outcomes following TEVAR: a control cohort, a failed TEVAR cohort, and a successful TEVAR cohort. The control cohort consists of 171 scans of non-pathologic aortas (SI Demographic Information). Failed repair is defined as reintervention or type 1A (proximal seal zone) endoleak in the follow-up period (using the most recent patient data available).

A balanced cohort of 18 patients with desired TEVAR outcomes and 18 patients with failed outcomes is analyzed. Patients all met the inclusion criteria of having both a pre-operative and post-operative CTA. A total of 45 scans (18 pre-operative, 27 post-operative) are included for

the desired outcomes group and 51 scans (18 pre-operative, 33 post-operative) are included for the failed outcomes group. Therefore, all outcomes analyses in this paper utilized a total of 267 scans (171 normal, 45 with desired outcomes, and 51 with failed outcomes). Outcomes are defined by the presence or lack of need for secondary surgical intervention more than 30 days from index TEVAR. Given the retrospective nature of this data cohort, all decisions to reintervene were made by the primary surgeon. Most common reasons for reintervention post-TEVAR were continued false lumen (FL) expansion and type 1A leak. Demographic, disease, and imaging information for this group is summarized in SI Demographic Information.

Computed tomography images were obtained via the Human Imaging Research Office (HIRO) at the University of Chicago Medicine, an institutional imaging research core that provided HIPPA-compliant, deidentified DICOM data for patients requested for this study. A variety of scanners were used to collect radiographic information. All data collection and analysis is performed in accordance with the guidelines established by the Declaration of Helsinki and under institutional review board approval (IRB20–0653 and IRB21–0299).

## Meshing algorithm and geometric parameterization

Three-dimensional aortic models are created from CTA image data using a custom workflow in Simpleware ScanIP (S-2021.06-SP1, Synopsys, Mountain View, CA). Aortic geometry is extracted from scans using a five-step algorithm which includes 1. segmentation, 2. noise reduction, 3. smoothing, 4. isolation of the segmentation outer surface, and 5. surface meshing. A representative schematic of the process is shown in Fig 2, and more information on the process can be found in SI Aortic Segmentation and Post-Processing from CTA Imaging. A triangular mesh for the outer surface is generated for each smoothed segmentation in ScanIP for analysis in Matlab (2021b, Mathworks, Natick, MA). A total of 15 meshed surfaces are generated for each segmentation (sampling 5 mesh densities and 3 smoothing variations), allowing for control of process-derived variance in surface curvature calculations.

Once a meshed geometry is created for each aorta, the Rusinkiewicz algorithm is used to calculate the per-vertex shape operator $\mathbb{S}_i = \begin{bmatrix} k_{1i} & 0 \\ 0 & k_{2i} \end{bmatrix}$, where $k_{1i}$ and $k_{2i}$ are the per-vertex principal curvatures. Briefly, the algorithm calculates the per-vertex shape operator as a weighted average of the shape operators of immediately adjacent faces. The per-face tensors are computed using a finite-difference approximation defined in terms of the directional derivative of the surface normal [46] (see SI Calculation of the Shape Operator for more details). Therefore, each vertex in the mesh is accompanied by a first principal curvature and second principal curvature that approximate the local shape at the intersection of neighboring faces. SI Artifact Removal outlines the artifact removal process that was implemented.

## Size and shape characterization

**Size.** Fig 3 shows our computational workflow for calculating aortic shape and size features. Aortic size is parameterized using the centerline length $\mathcal{L}$ and aortic radius $\ell$. While $\mathcal{L}$ is singular for each aorta, $\ell$ is a distribution, especially in diseased aortas with heterogeneous shapes, and therefore a family of radii. Multiple measured values can be used to represent this distribution to characterize aortic size: mean aortic radius ($\langle R \rangle$), median aortic radius ($\tilde{R}_2$), and maximum aortic diameter $2R_m$. $\ell$ can also be calculated directly from surface curvatures. The mean Frobenious norm of the per-vertex shape operator $\mathbb{S}_i$, $C_i = \frac{1}{2}(k_{1i}^2 + k_{2i}^2)$, is the per-vertex Casorati curvature [47–49]. Averaging over the entire aorta gives the mean Casorati

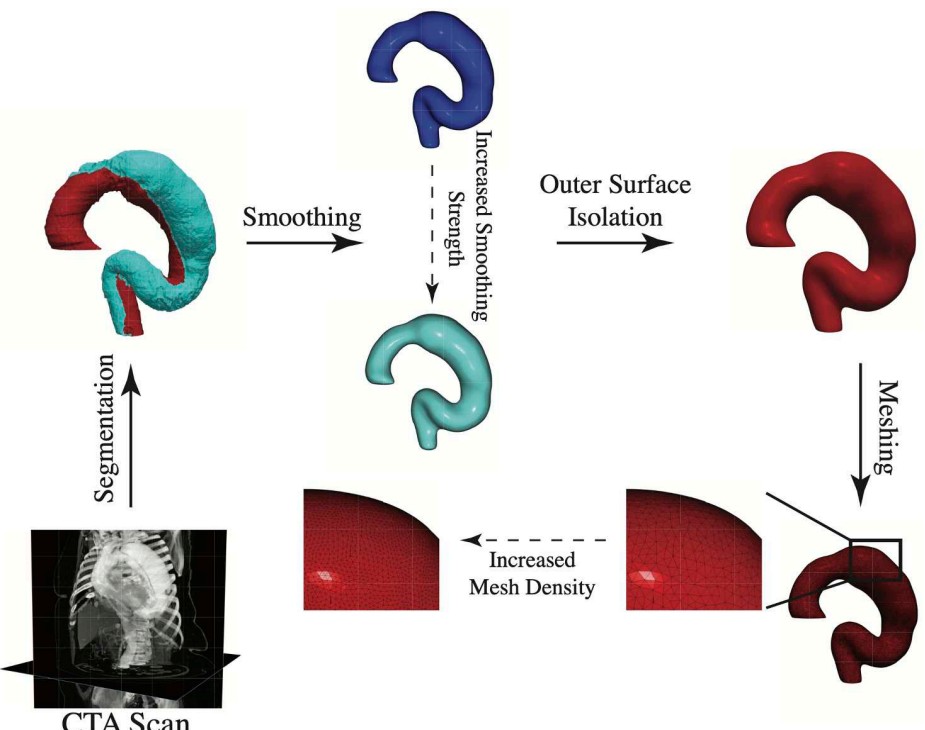

**Fig 2. Image Processing Workflow.** Aortas are segmented from CTA imaging scans of the chest, followed by smoothing of the segmentation, isolation of the segmentation outer surface, and triangular surface meshing. The noise reduction procedure encompasses the smoothing and meshing steps, in which multiple smoothing parameters and mesh density variations generate multiple plausible surface meshes representing the segmentation.

curvature,

$$\langle C^{1/2} \rangle = \frac{1}{A_T} \sum_{j=1}^{k} \langle C_j^{1/2} \rangle A_j \qquad (1)$$

where $\langle C_j^{1/2} \rangle = \frac{1}{p} \sum_{i=1}^{p} \sqrt{C_i}$ is the per-partition mean, $k$ is the number of partitions, $A_j$ is the area of a partition, and $p$ the number of vertices within each partition. The need for dividing the surface into $k$ smaller partitions is not necessary to obtain accurate mean Casorati curvatures; however as explained below, it becomes necessary when performing shape calculations. We keep it here from an implementation consistency standpoint because $C_i$ is calculated from principal curvatures. The inverse mean Casorati curvature $\langle C^{1/2} \rangle^{-1}$ is a measure of aortic size along the direction of greatest curvature, which is the aortic radius. Lastly, higher-order functions of size such as total aortic area, $A_T$, and aortic volume, $V$, are calculated directly from the surface segmentations. Total aortic area is also calculated from the sum of per-element areas: $A_T = \sum_{i=1}^{g} a_i$, where $g$ is the total number of triangular mesh elements on a given aortic surface. It is important to note that only $\langle C^{1/2} \rangle^{-1}$ allows for a measure of aortic size using only information contained in $\mathbb{S}_i$; all other size measures necessitate additional information.

**Shape.**   The size-independent parameterization of shape is obtained through an effective mapping of normals on the external aortic surface $\mathcal{S}$ to the unit sphere $S^2$. Operationally, we do not directly perform the Gauss mapping. Instead, we calculate the total curvature over local

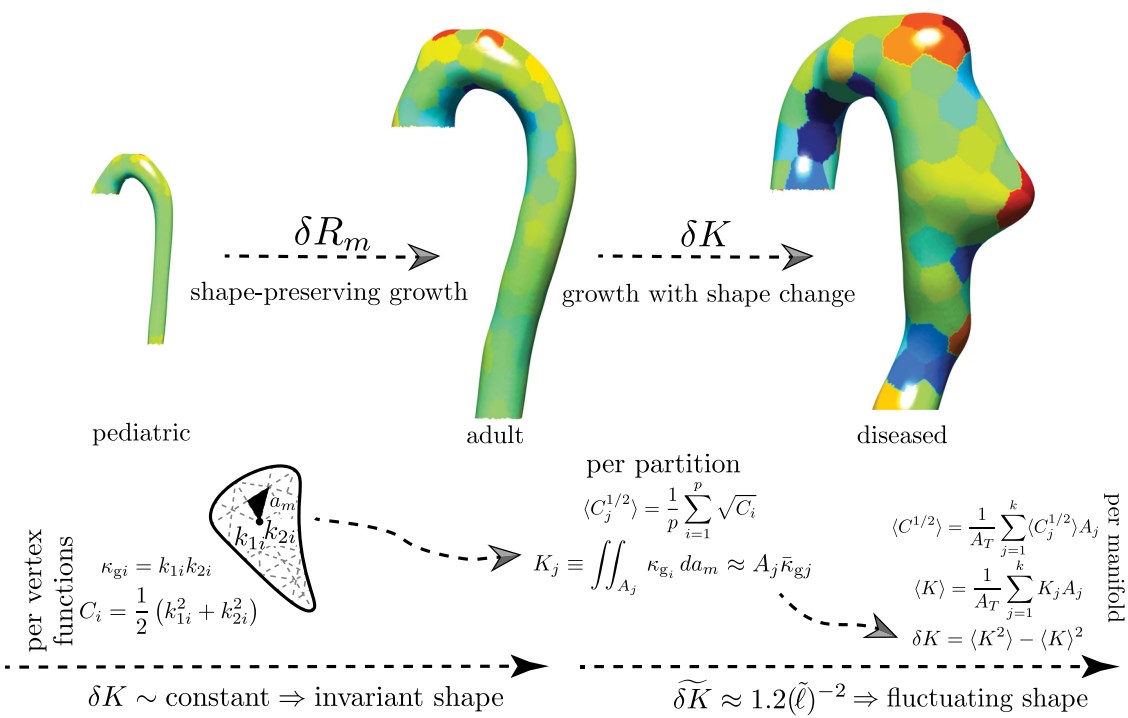

**Fig 3. Multi-Scale Surface Curvature Calculations.** By mapping the aortic surface to the unit sphere (Gauss map) [45], we have an independent measure of shape. The per-vertex shape operator $\mathbb{S}_i$ is calculated using the Rusinkiewicz algorithm [46]. To minimize noise, the aorta is divided into multiple partitions with area $A_j$. The local integrated Gaussian curvature $K_j$ is calculated as the product of each partition area and mean Gaussian curvature, $\bar{\kappa}_{gj}$. $K_j$ is equivalent to the signed partition area $\tilde{A}_j$ mapped out by the normals projected onto the unit sphere. We define aortic shape by studying the statistics of the distributions of $K_j$. $\langle K \rangle$ and $\delta K$ are the first and second distribution moments that define aortic shape geometry, respectively.

regions (partitions) of the aortic surface with area $A_j$, which by the local Gauss-Bonnet theorem is the holonomy angle of the circumnavigated area, a topologic quantity [47]. In our analysis, given geometrically highly heterogenous shapes, a meshed surface is employed and the per-vertex Gaussian curvature is extrinsically calculated from the shape operator: $\kappa_{gi} = |\mathbb{S}_i| = k_{1i}k_{2i}$. The area elements are intrinsic to the surface and $\iint_A dA = \sum_{i=1}^{p} a_i = A_j$; however, since a generic triangular mesh is used, no information about the surface metric exists. Knowledge of the metric would be needed to carry out the integration of per-vertex Gaussian curvature explicitly. Our computational approach therefore relies on splitting the integration into a product $K_j \approx A_j\bar{\kappa}_{gj}$, where $\bar{\kappa}_{gj} = \frac{1}{p}\sum_{i=1}^{p}\kappa_{gi}$. To bring the mean Gaussian curvature within a partition outside the integral, $\kappa_{gi}$ must vary weakly within $A_j$. Clearly, this variance is minimized when $A_j = a_i$. As shown in SI Sensitivity to Partition Size, calculating $K$ at this scale holds little information about overall aortic shape. To understand the necessity of sub-dividing the aortic surfaces into partitions where $a_j < A_j < A_T$, the scale space nature of the problem needs to be explored and appreciated. Since the per-vertex Gaussian curvature is calculated on a mesh with an average element spacing proportional to the CT scan resolution ($\sim \mathcal{O}(0.5)$ mm); this value assumes the mesh-imposed inner scale. However, as is appreciated by examining the evolution in Fig 1, the change in aortic shape, the increasing 'bumpiness', is occurring on a much larger length scale. Visual inspection of the shapes shows this scale to be on the order of the aortic radius or $\ell$. The mesh-imposed inner scale is $\sim \ell/1000$. As discussed

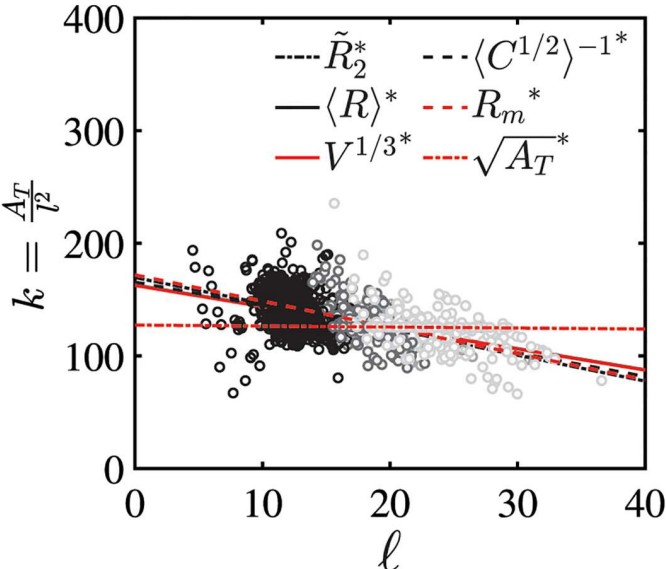

**Fig 4. Number of Surface Partitions Imposed by the Inner Scale $\ell$.** Data for 302 aortas, including non-pathologic (black circles), pathologic with failed TEVAR (light gray circles), and pathologic with successful TEVAR (dark gray circles) aortas are plotted. The linear scaling can be used to define $A_j \sim \ell^2$, which sets the number of partitions $k$ used in the Gauss map calculations. The various linear fits are taken for different definitions of size: maximum aortic diameter ($2R_m$, red dashed line), mean radius ($\langle R \rangle$, black solid line), median radius ($\tilde{R}_2$, black dotted line), and mean inverse linearized aortic Casorati curvature ($\langle C^{1/2} \rangle^{-1}$, black dashed line) are equivalent. Dimensionally scaled, aortic area ($\sqrt{A_T}$, red dotted line) and volume ($V^{1/3}$, red solid line) are also linear when plotted against $\ell = 2R_m$. In this case, the fits are normalized by the pre-factors obtained from their fitting to the maximum dimeter (Fig 5). The normalized data is shown to demonstrate that $k$ is independent of the specific size measure used to set the inner scale $\ell$.

in SI Aortic Segmentation and Post-Processing from CTA Imaging, the mesh resolution was set by the inherent resolution of the CT data and by the need of a finely enough spaced computational mesh to calculate discrete derivatives of surface normals. However, from an aortic shape classification standpoint, using the mesh-imposed inner scale runs the risk of imposing a large amount of "spurious resolution" or "false detail", commonly seen with scale space problems when an inappropriately small inner scale is selected [47]. We impose $\ell$ as the inner-scale by sub-dividing the aortic surface into $k$ area partitions via a Voronoi decomposition using the $k$-means algorithm, in which $k = A_T/\ell^2$ (see Fig 4 which shows that $k$ is independent of how $\ell$ is measured and relatively constant with respect to $\ell$). For subsequent analysis, $\ell = \tilde{R}_2$ (indicating the median $R_2$, where $R_2 = 1/k_{2i}$ is calculated per-vertex from $\mathbb{S}_i$). We use the $k$-means++ algorithm to find initial centroid seeds, and the $k$-means algorithm is performed with a maximum of 10000 iterations. SI Sensitivity to Partition Size shows how $k$ can vary by a factor of 10 and does not impact the results discussed in the following sections. Lastly, SI Jensen-Shannon Divergence of Within-Partition Gaussian Curvature outlines our use of the Jensen-Shannon Divergence to check that at this inner-scale Gaussian curvatures still show acceptably small variance, allowing us to bring the mean Gaussian curvature within a partition out of the area integral and replace the area integral with the sum of mesh element areas.

Having partitioned the aortic surface into $k$ partitions of size $\sim \ell^2$, the per partition total curvature

$$K_j = \sum_{m=1}^{q} a_m \left( \frac{1}{p} \sum_{i=1}^{p} \kappa_{gi} \right) = A_j \bar{\kappa}_{gj} \tag{2}$$

is calculated. To characterize the entire aorta, the sum of per partition total curvatures (a topologic invariant)

$$\sum K = \sum_{j=1}^{k} K_j \tag{3}$$

and the fluctuations in total curvature (a normalized measure of shape) across the entire manifold surface are calculated:

$$\delta K = \frac{1}{A_T} \sum_{j=1}^{k} K_j^2 A_j - \left( \frac{1}{A_T} \sum_{j=1}^{k} K_j A_j \right)^2 \tag{4}$$

The mean Gaussian curvature

$$\langle \kappa_g \rangle = \frac{1}{A_T} \sum_{j=1}^{k} \langle \kappa_{g_j} \rangle A_j \tag{5}$$

and the fluctuation in Gaussian curvature of the manifold

$$\delta \kappa_g = \frac{1}{A_T} \sum_{j=1}^{k} \kappa_{g_j}^2 A_j - \left( \frac{1}{A_T} \sum_{j=1}^{k} \kappa_{g_j} A_j \right)^2 \tag{6}$$

are also calculated.

**Statistics.** Each aorta contains $k$ partitions. All calculations are performed for each of the 15 meshed models per aorta. Of note, the $k$-means partitioning is independently performed for each mesh. An average value for $\langle C^{1/2} \rangle$, $\Sigma K$, $\delta K$, $\langle \kappa_g \rangle$, and $\delta \kappa_g$ is computed for the 15 meshes, such that each scan becomes represented by a single value that aggregates the variation from the smoothing and meshing algorithms. To quantify variability from the partitioning procedure itself, the entire process (partitioning, partition-level curvature calculations, manifold-level curvature calculation, and averaging between meshes) is repeated 10 times to obtain 10 replicates of each value per aorta. The mean of these replicates is reported and a sensitivity analysis is performed in SI Sensitivity to Partition Size. The error bars in all data plotted in the two-dimensional feature space represent ±1 standard deviation to quantify the variability in the results from the partitioning replicates.

## Aortic feature space classification using machine learning

The classification accuracy of different shape and size metrics in determining aortic disease states (normal aortas, successful TEVAR, and failed TEVAR) is calculated using the means of the distributions, the mean of the means of the distributions, and a logistic regression classification. The distribution mean-based methods model is reflective of current clinical practice, in which clinicians are most cognizant of the characteristics of a "typical" patient in each group (the mean) and are less aware of the variation within each group (the standard deviation). The first model defines each threshold $t$ as the mean value of the parameter for the two neighboring distributions. After these boundaries are defined, each scan is assigned a "predicted" classification according to its geometric characteristic, and the accuracy of the predictions is computed. The second model defines each threshold as the mean value of the means of the two neighboring distributions.

A multinomial logistic regression with 1000 random permutations of train-test splits with a 50% training and 50% testing distribution is used for the third model. Logistic regression is a

classification method that models $p(X) = Pr(Y = 1|X)$, the probability that some response variable $Y$ takes on a specific value of 1 (a binary classification) based on the input data $X$, using the logistic function: $p(X) = \frac{e^{\beta_0+\beta_1 X}}{1+e^{\beta_0+\beta_1 X}}$. The parameters $\beta_0$ and $\beta_1$ are estimated from training data using the maximum likelihood method. The accuracy of this model in predicting binary outcome $Y$ is determined by subdividing the dataset into a training set and a testing set, fitting $\beta_0$ and $\beta_1$ using the training set, and computing the error rate between model predictions and data for the testing set.

As the classification problem is between three classes—non-pathologic aortas, diseased aortas with desired outcomes following TEVAR, and diseased aortas with failed outcomes following TEVAR—multinomial logistic regression is used. Multinomial logistic regression is an extension of logistic regression for the setting of $H > 2$ classes with coefficient estimates defined as follows:

$$Pr(Y = k|X = x) = \frac{e^{\beta_{k0}+\beta_{k1}x_1+\cdots+\beta_{kp}x_p}}{1 + \sum_{l=1}^{H-1} e^{\beta_{l0}+\beta_{l1}x_1+\cdots+\beta_{lp}x_p}} \tag{7}$$

The variability of the logistic decision boundary includes lasso regularization applied to $\langle C^{1/2} \rangle$ to examine the impact of increasing the model's dependence on shape over size. The objective function is the penalized negative binomial log-likelihood:

$$\min_{(\beta_0,\beta)\in\mathbb{R}^{p+1}} \left( -\left[ \frac{1}{N}\sum_{i=1}^{N} y_i \cdot (\beta_0 + x_i^T\beta) - \log(1 + e^{(\beta_0+x_i^T\beta)}) \right] + \lambda||\beta||_1 \right) \tag{8}$$

Two decision boundaries are created using two binomial logistic regression classifiers. The hyperparameter $\lambda$ is varied to understand the impact of decreasing the influence of size on clinical decision-making to maximize the impact of using a shape-based classifier [50, 51].

## Finite element simulations

The utility of $K$ over $\kappa_g$ is investigated in a more controlled setting by simulating local growth in ideal geometries and examining the associated change in $\kappa_g$ versus $K$. Two geometries are tested: a sphere that experiences a small change in global size followed by surface deformation due to local growth and an idealized aorta that experiences a larger change in global size followed by surface deformation due to local growth. Growth is modeled using a morphoelastic model in ABAQUS (2018, Dassault Systèmes, Waltham, MA) that decomposes the deformation gradient $F$ into an elastic contribution $F_e$ and a growth contribution $F_g$: $F = F_e F_g$ [52–54]. Adopting the assumption that $F_g$ does not explicitly contribute to the free energy in previous work on computational modeling of multiplicative growth [52–54], a neo-Hookean (NH) strain energy function of the following form is used: $W = W(F, F_g) = W_e(F_e)$. When there is no growth, $F_g = I$ and $F_e = F$, which represents a purely elastic deformation. With this decomposition and the NH strain energy, the stress can be derived and updated during the loading process according to the following relations:

$$S_e = 2\frac{\partial W_e}{\partial C_e} \tag{9}$$

$$\sigma = \frac{1}{J} F_e S_e F_e^T \tag{10}$$

Where $S_e$ and $\sigma$ are the second Piola-Kirchhoff stress and Cauchy stress, respectively. The above model is implemented with ABAQUS Explicit solver and using the VUMAT subroutine

[55, 56]. Assuming isotropic growth with constant growth rate in the longitudinal-circumferential plane, the growth contribution $F_g$ becomes:

$$F_g = \begin{bmatrix} 1 & 0 & 0 \\ 0 & v & 0 \\ 0 & 0 & v \end{bmatrix} \qquad (11)$$

where $v$ is the growth factor and the growth rate is $\dot{v}$. See SI Finite Element Simulations for further details.

## Results

Three major and two minor results come from our analysis. The first major result is that aortic size scales as a generalized bent cylinder and depends only on a single length scale, $\ell$. The second major result is that projecting aortas into a two-dimensional space defined by $(\delta K, \ell^{-1})$ separates aortic geometries into shape-preserving self-similar growth and growth with shape changes; moreover, all aortic geometries are shown to be topologically homeomorphic to $T^2$. The third major result shows that the $(\delta K, \ell^{-1})$-feature space outperforms all other size and shape measures when applied to the predictive classification of TEVAR success. The first minor result shows that the data projected into the normalized $(\widetilde{\delta K}, \tilde{\ell}^{-1})$ space are best fit to a simple power law ($\widetilde{\delta K} \sim \tilde{\ell}^{-2}$). The second minor result shows that $\delta K$ and $\delta \kappa_g$ hold similar information, however, $\delta K$ provides a stronger signal of shape changes when shape and size changes occur simultaneously.

### Universal scaling of aortic size

The analysis includes a variety of size measures previously described in the literature, including the traditional metric of maximum diameter ($2R_m$) and higher-dimensional values of area ($A_T$) or volume ($V$) (Fig 5). When each length scalar is plotted versus maximum diameter, the data collapses onto lines, with linear size measures having a slope of 1 and higher-dimensional measures ($\sqrt{A_T}$ and $V^{1/3}$) having slopes that correspond to modeling aortic size as a generalized bent cylinder. In generalized bent cylinders, area and volume scale as $A_T \sim 2\pi\ell \times \mathcal{L}$ and $V \sim \pi\ell^2 \times \mathcal{L}$. Hypothesizing a linear relationship between axial length $\mathcal{L}$ and cross-sectional circular radius $\ell$, $\mathcal{L} \sim c\ell$, the reparameterized equations $A_T \sim 2\pi c\ell^2$ and $V \sim \pi c\ell^3$ are obtained.

Fig 5 shows that $\sqrt{A_T} \sim 10\ell$ and $V^{1/3} \sim 3.5\ell$, which implies that $A_T \sim 100\ell^2$ and $V \sim 42.88\ell^3$. Solving for $c$, we obtain $c \sim 15.9$ for area and $c \sim 13.6$ for volume. The pre-factor $c$ can alternatively be calculated using $\mathcal{L} \sim c\ell$ directly. $\mathcal{L}$ is best approximated using the length of the aortic centerline measured from the segmentations. Thus, $c \sim \frac{\mathcal{L}}{\ell}$ can be calculated on each aortic geometry (Fig 6), with $c = 16.6 \pm 2.4$ when $\ell \sim \langle C^{1/2}\rangle^{-1}$ is used. The values of $c \sim 15.0$, $c \sim 13.6$ and $c \sim 16.6$ indicate similar constants that are produced using independent methods and that quantify a linear relationship between aortic axial length and cross-sectional size. This demonstrates that aortic geometry follows a universal linear scaling between different size measures and globally retains an invariant toroidal-cylindrical geometry. This indicates that additional information is unlikely to exist with higher-dimensional size measurements that have been the focus of recent literature and provides geometric reasoning behind the well-known utility of maximum diameter in aortic management.

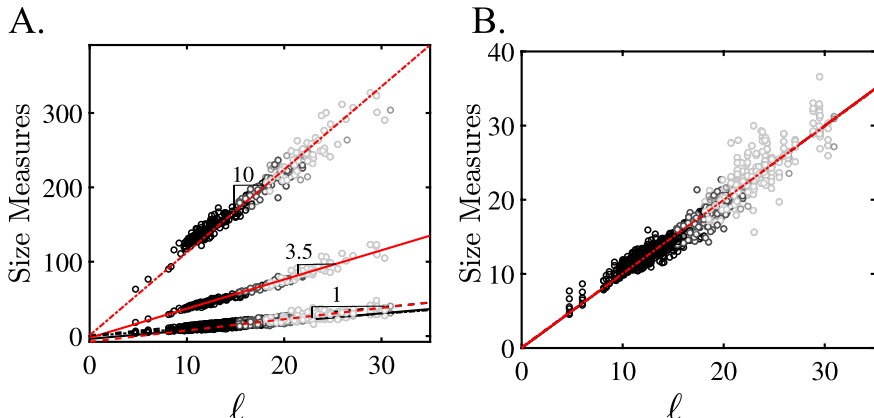

**Fig 5. Universal Scaling of Aortic Size.** Data for 302 aortas, including non-pathologic (black circles), pathologic with failed TEVAR (light gray circles), and pathologic with successful TEVAR (dark gray circles) aortas are plotted. A. shows that parameterizations of aortic size (mm) including maximum aortic diameter ($2R_m$, red dashed line), mean radius ($\langle R \rangle$, black solid line), median radius ($\tilde{R}_2$, black dotted line), and mean inverse linearized aortic Casorati curvature ($\langle C^{1/2} \rangle^{-1}$, black dashed line) are equivalent. Dimensionally scaled, aortic area ($\sqrt{A_T}$, red dotted line) and volume ($V^{1/3}$, red solid line) are also linear when plotted against $\ell = 2R_m$. All size measures can be collapsed onto a single master curve (B.), proving that all aortas scale as generalized bent cylinders parameterizable by a single length scale $\ell$.

## Role of shape deviation

Fig 7 plots the data from 302 aortic segmentations into two spaces: $(\sum K, \tilde{\ell}^{-1})$ and $(\widetilde{\delta K}, \tilde{\ell}^{-1})$. Size is represented in both by $\tilde{\ell}^{-1} = \ell^{-1} / \langle \ell \rangle_{norm}^{-1}$, where $\langle \ell \rangle_{norm}$ is the mean of the normal aortas. As was shown above, the different measures of $\ell$ differ only by a constant pre-factor. Therefore, the same normalized size axis is obtained irrespective of how $\ell$ is measured. $\sum K$ is

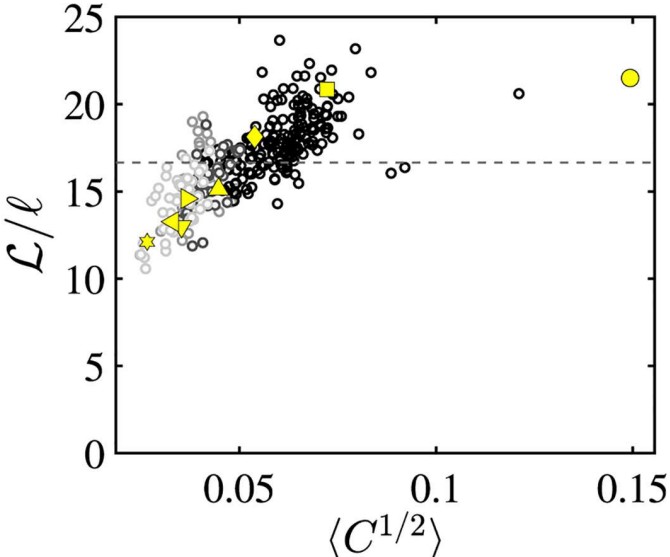

**Fig 6. Length-to-Size Ratio as Function of Size.** Ratio of centerline length $\mathcal{L}$ to radial size $\ell$. For the relationship $\mathcal{L} \sim c\ell$ indicating a linear scaling between axial length and cross-sectional circular radius, we obtain $c = 16.6 \pm 2.4$. The yellow symbols indicate selected aortas shown in Fig 7A.

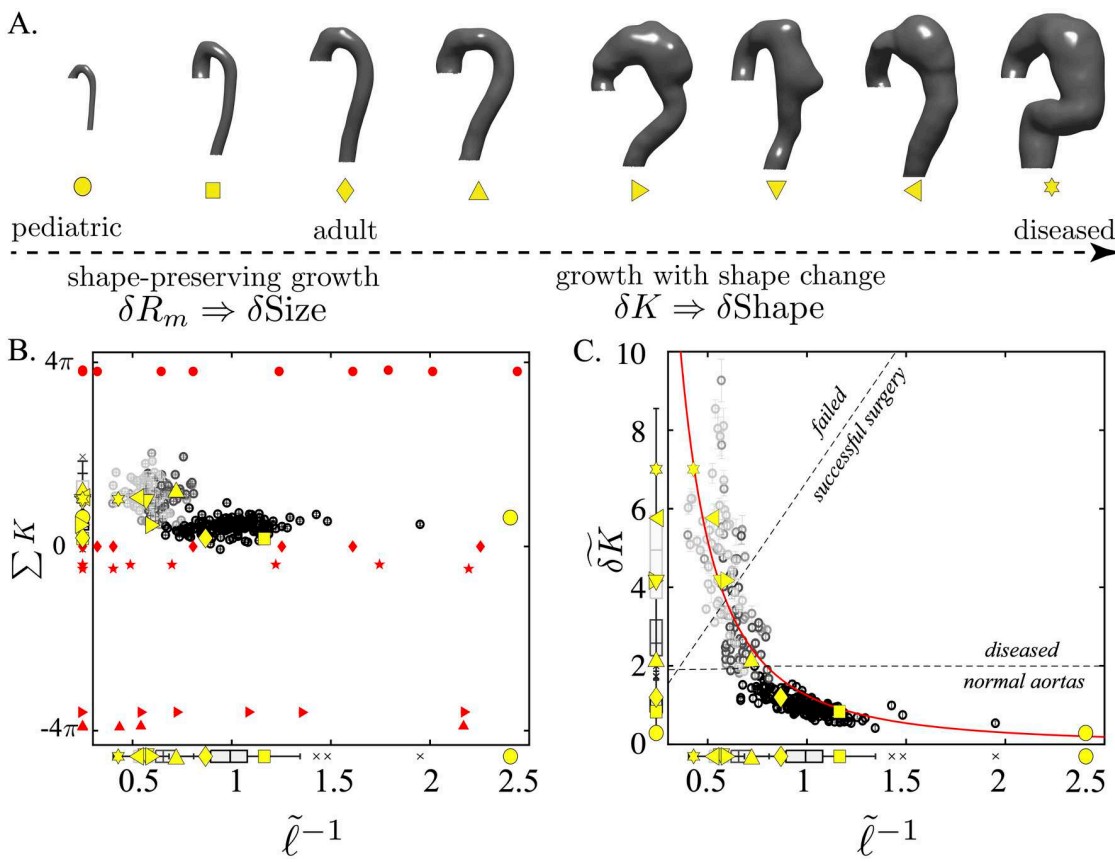

**Fig 7. Aortic Topological Invariance and Aortic Clustering in $(\widetilde{\delta K}, \tilde{\ell}^{-1})$-space.** A. The eight canonical representative aortas along the normal-to-diseased axis (left to right): a 3-year-old child, healthy adults, and type B aortic dissection (TBAD) patients at varying degrees of aneurysmal degeneration. Two clinical regimes exist: shape preserving growth and growth with shape changes. B. shows the topologic equivalence of all aortic shapes to tori (red stars) and cylinders (red diamonds); the yellow symbols correspond to specific aortic shapes along the normal-to-diseased axis (A.). Red circles correspond to perfect spheres of varying size; pseudospheres and catenoids are depicted as red rightward-pointing triangles and upward-pointing triangles, respectively. C. shows the optimal two-dimensional aortic geometric feature space with independent axes for size and shape. The solid red curve $\widetilde{\delta K} = 1.2\tilde{\ell}^{-2}$ is a best fit to the data. The power-like behavior is further supported by the probability distribution of $\delta K$ (Fig 8). The aortas separate into shape invariant (normal) and shape fluctuating (diseased) populations. Furthermore, this feature space defines decision boundaries that correctly classify diseased patients based on success of TEVAR.

the total integrated Gaussian curvature of each aorta and mathematically defines its topology. Fig 7B plots the aortic data points (black and grey circles) along with the representative aortas in yellow. Red points are calculated on analytically generated surfaces (see SI Ideal Shapes) analyzed through the same analysis pipeline as the aortic data: spheres (red circles), tori (red stars), cylinders (red diamonds), pseudospheres (red rightward pointing triangle), and catenoids (red upward pointing triangle) of various sizes.

These ideal shapes represent the three canonical geometries: spherical, parabolic, and hyperbolic. Their total curvature values of $\Sigma K = 4\pi$ (spherical), $\Sigma K = 0$ (parabolic or Euclidian), and $\Sigma K = -4\pi$ (hyperbolic) are in agreement with analytical results from differential geometry [57]. This agreement for the ideal surfaces is an important validation of the computational methodology used in this paper. The aortic data clearly align along the $\Sigma K = 0$ line and cluster with the tori and cylinders. The normal aortas (solid black dots) show a very tight distribution with very little variance along $\Sigma K$. The diseased aortas show more spread, which is likely a

consequence of the complex shapes encountered in the group. Nevertheless, what is striking in this data is that throughout the morphologic life span of an aorta (during normal growth and diseased degeneration), any given aorta is topologically equivalent to a generalized bent cylinder or torus homeomorphic to $T^2$. Consequently, by the Gauss-Bonnet Theorem, aortic topologic invariance implies that as aortas deform, be it through normal growth or pathology, every increase in $K$ somewhere must be balanced by a proportional decrease elsewhere on the aortic surface [45, 57, 58]. Therefore, the distribution of $K$ across the aortic surface should hold information about shape. Since $\Sigma K$ is constant, the variance $\delta K = \langle K^2 \rangle - \langle K \rangle^2$ captures the balance of positive and negative curvature regions across the surface and is a measure of shape deformation. Normalized $\delta K$ is defined as $\widetilde{\delta K} = \delta K / \langle \delta K \rangle_{norm}$, where $\langle \delta K \rangle_{norm}$ is the mean $\delta K$ of the normal aortas. Examination of the data in Fig 7C shows that $\widetilde{\delta K}$ quantitatively captures the above-mentioned qualitative descriptions of aortic shape. Our three aortic populations naturally separate into three regions in $(\widetilde{\delta K}, \tilde{\ell}^{-1})$-space. In the asymptotic limit $\widetilde{\delta K} \to 0$, self-similar aortic growth occurs; all normal aortas exist along this limit. In the other limit, $\widetilde{\delta K} \to \infty$, fluctuations in shape become seemingly independent of size change. The most distorted and convoluted aortas, which also correspond to patients with failed endovascular surgeries, cluster in this limit. The two limits are joined through a transition region that appears as an elbow in the two-dimensional shape-size feature space. Interestingly, the dissection patients who had successful surgery predominantly cluster in this region. The utilization of this novel $(\widetilde{\delta K}, \tilde{\ell}^{-1})$-space for aortic classification is further developed below.

## Power law fit in the $(\widetilde{\delta K}, \tilde{\ell}^{-1})$ feature space

In the $(\widetilde{\delta K}, \tilde{\ell}^{-1})$ geometric feature space, the data can be fit to a power law ($\widetilde{\delta K} \sim \tilde{\ell}^{-2}$). Here we follow an established statistical framework of discerning power-law behavior in empirical data to determine whether the data is truly consistent with a power-law [59]. Fig 8 shows separate probability distributions for $\delta K$ and $R_m$ and then plots these on doubly logarithmic axes. Logarithmic distributions, $p(x) \propto x^{-a}$, are linear in log-log space: $\ln p(x) = \alpha \ln x + c$, where $c$ is a constant. The log-log transformed $\delta K$ distribution suitably fits a straight line with $R^2 \approx 0.74$, while the $R_m$ data does not fit a line ($R^2 \approx 0.002$). This indicates that a power law may be an appropriate fit for the shape metric $\delta K$ and is not an appropriate fit for the size parameter $R_m$. The size data are well approximated by a Gaussian distribution, a fact demonstrated in the literature on aortic size distributions [30].

## Classification with predictive modeling

The strong correlation between $\delta K$ divergence and the evolution of aortic pathology begs application to treatment planning. Fig 9 compares the effectiveness of aortic size, the clinical standard, with the $(\widetilde{\delta K}, \tilde{\ell}^{-1})$ shape-size feature space in determining aortic disease states (normal aortas, successful TEVAR, and failed TEVAR). Significant variability in aortic size within the broader patient population is captured by this single institutional dataset. Furthermore, there is overlap between the normal and diseased populations and as such, classification accuracy dramatically changes due to slight differences in model parameterizations (from 73% in Fig 9A, 83.9% in Fig 9B, and 87.0±2.3% in Fig 9C). The dual parameterization of both aortic size and shape—two orthogonal metrics—expands the geometric characterization into a two-dimensional space (Fig 9D and 9E). Classification accuracy increases to 90.3% in Fig 9D and 92.8±1.7% in Fig 9E.

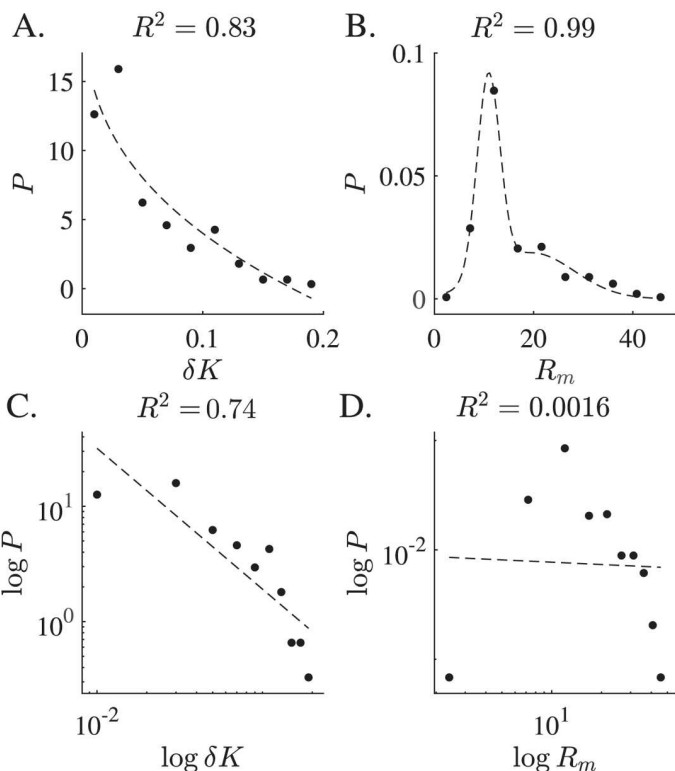

**Fig 8. $\widetilde{\delta K}$ Fits Power Law Distribution while Size is Gaussian.** Probability distributions of $\delta K$ and $R_m$ are plotted. A. The $\delta K$ distribution is fitted to a power law in the form $P = ax^b + c$. C. A linear fit $\log P = b \log \delta K + c$ achieves a high $R^2$. B. The $R_m$ distribution is well-fit to a two-term Gaussian in the form $P = a_1 e^{\left(-\frac{x-b_1}{c_1}\right)^2} + a_2 e^{\left(-\frac{x-b_2}{c_2}\right)^2}$. D. When a linear fit is applied to the log-transformed data, $\log P = b \log R_m + c$, a low $R^2$ value results.

As shown in Fig 10A, there is no significant difference amongst size predictors such as aortic volume, surface area, median and maximum diameters shown in Fig 5, as well as Gaussian-curvature based size measures like the L2-norm of the Gaussian curvature (GLN) and the area-averaged Gaussian curvature (GAA) [40, 60, 61], see SI Other Shape Metrics. There is a major difference with shape measures: clinical shape measures, which are based on the aortic centerline and include tortuosity index [21, 43, 44, 62], question mark angle [44, 63], cross-sectional eccentricity [42, 64], and mean centerline curvature [43], significantly underperform $\delta K$ in aortic disease state classification (Fig 10B). Similarly, $\delta K$ outperforms other shape measures described in the biomedical engineering literature including the sphericity index ($\chi$) [65], flatness index ($\gamma$) [65], Gaussian curvature ($\kappa_g$) [40, 41], area-averaged mean curvature (MAA), and L2-norm of the mean curvature (MLN) [40, 60, 61] (Fig 10C).

## Shape evolution modeling

In this paper, $K$ is shown to be a topologic invariant across aortic anatomies and $\delta K$ a strong function of aortic shape changes. The integrated total curvature $\Sigma K$ is a topologic invariant and remains approximately constant across all aortic anatomies (Fig 7B). $\delta K$ is approximately constant across the range of normal aortas, matching empirical knowledge that non-pathologic aortas experience shape-preserving growth. $\delta K$ diverges with rapidly degenerating aortas, especially those where TEVAR ultimately failed. The role of $\delta K$ in this divergence is elucidated

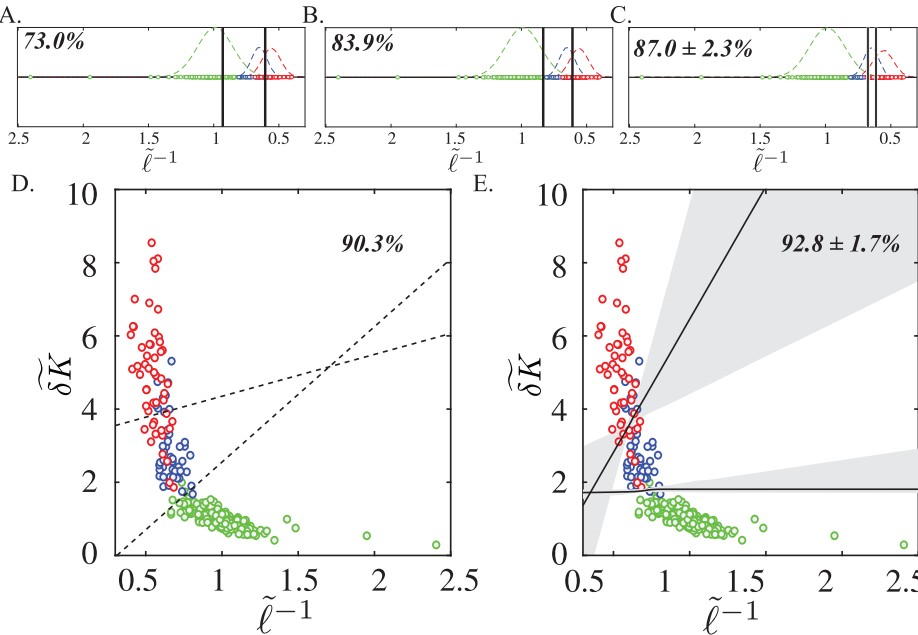

**Fig 9. Clustering Analysis in $(\widetilde{\delta K}, \tilde{\ell}^{-1})$ Geometric Feature Space Shows Superior Accuracy and Stability Compared to Size Alone.** The $(\widetilde{\delta K}, \tilde{\ell}^{-1})$ geometric feature space improves upon current sized-based methods. The clinical paradigm relies on size metrics alone to classify aortic disease states (green for normal aortas, blue for successful TEVAR, and red for failed TEVAR). However, broad within-group size distributions indicate considerable variability in aortic sizes within the general population. Clinicians routinely utilize statistical means of these distributions as thresholds for classifying disease states, but linear decision boundaries are highly sensitive to small changes in model setup. A. A 73.0% accuracy for classifying the 3 states is obtained when each threshold is defined as the mean $\langle C^{1/2} \rangle$ of the two neighboring distributions. B. An 83.9% accuracy is achieved when the threshold is defined as the midpoint of the means of individual class distributions. C. An 87.0% accuracy is obtained when a logistic regression classifier is used. Thus, small changes in how a threshold is defined dramatically alter the perceived utility of size. D. The $(\widetilde{\delta K}, \tilde{\ell}^{-1})$ shape and size-based geometric feature space allows for the utilization of two independent parameters to characterize aortic disease state. A 90.3% classification accuracy is obtained when defining thresholds according to the mean $\delta K$ and $\langle C^{1/2} \rangle$ of each patient group. E. A 92.8% mean accuracy with a standard deviation of only 1.7% is obtained using a logistic regression classifier with varying regularization parameters. The shaded region indicates the interquartile range of decision boundaries and demonstrates the robustness of the two-parameter space. Unlike the single parameter space, the presence of two physically interpretable and orthogonal asymptotic limits ensures more effective classification.

with FE simulations on spheres (Fig 11) and an idealized aorta (Fig 12). More information on the simulation parameterizations can be found in SI Finite Element Simulations.

Fig 11 shows how for the case of a small change in global size, as exhibited by the narrow range of the x-axis (9.5%), the fluctuation in Gaussian curvature ($\delta\kappa_g$) accurately captures surface deformation just as well as the fluctuation in total curvature ($\delta K$). This is to be expected as the main function of total curvature is to normalize size effects by mapping the aortic surface to $S^2$. Thus, with little change in overall size, Gaussian curvature remains an effective indicator of shape. This is to be contrasted with what happens when size does change, as in the real patient data (400%) or the idealized aorta shown in Fig 12 (50%). Gaussian curvature is no longer an effective indicator of shape when size also changes because it convolutes shape and size effects. On the other hand, total curvature normalizes size effects and effectively captures the degeneration in the shape of the ideal aorta. This experimental evidence in a controlled simulated system bolsters the empirical data and mathematical formulation, proving the significance of $\delta K$.

## Discussion

Type B aortic dissection (TBAD) is a life-threatening disease with significant associated morbidity and mortality [17, 19, 20]. While the old paradigm of open surgical repair was fraught with peri-operative risk, new minimally invasive approaches like TEVAR often trade a decrease in initial operative risk for a higher risk of long-term repair failure. Proper identification of patients for TEVAR is therefore critical and necessitates the definition of an appropriate classification scheme [17, 22, 66, 67]. While previous work has focused on linking changes in aortic anatomy and suitability for repair, there remains a dire need to improve our understanding of how best to define geometric changes and to understand their impact on patient outcomes.

Projection of aortic anatomy into the $(\delta K, \ell^{-1})$-space provides an improved ability to differentiate aortas along the entire spectrum of growth and pathology, including both normal size-related development and pathologic shape-related changes that occur secondary to aortic dissection. We demonstrate that in normal conditions, the aorta undergoes shape-invariant growth (see Fig 7), while, in diseased states, the aorta experiences shape fluctuations defined by increasing $\delta K$. As shown in Fig 10, $\delta K$ significantly outperforms all other available measures of shape in predicting clinical treatment outcomes, including tortuosity, which is prevalent in the clinical arena.

Furthermore, because of the invariant global cylindrical geometry of the aorta, the parameterization of size is dependent only on the single length scale $\ell^{-1}$. Higher-dimensional characterizations of size, including area $A_T$ and volume $V$, do not provide additional information [68, 69]. Thus, current efforts to replace $2R_m$ with area or volume [68, 69] are unlikely to yield substantially more information (Figs 5 and 10). This universal size scaling provides the quantitative basis behind the utility of maximum diameter throughout the decades of aortic management and further validates the study of shape [67].

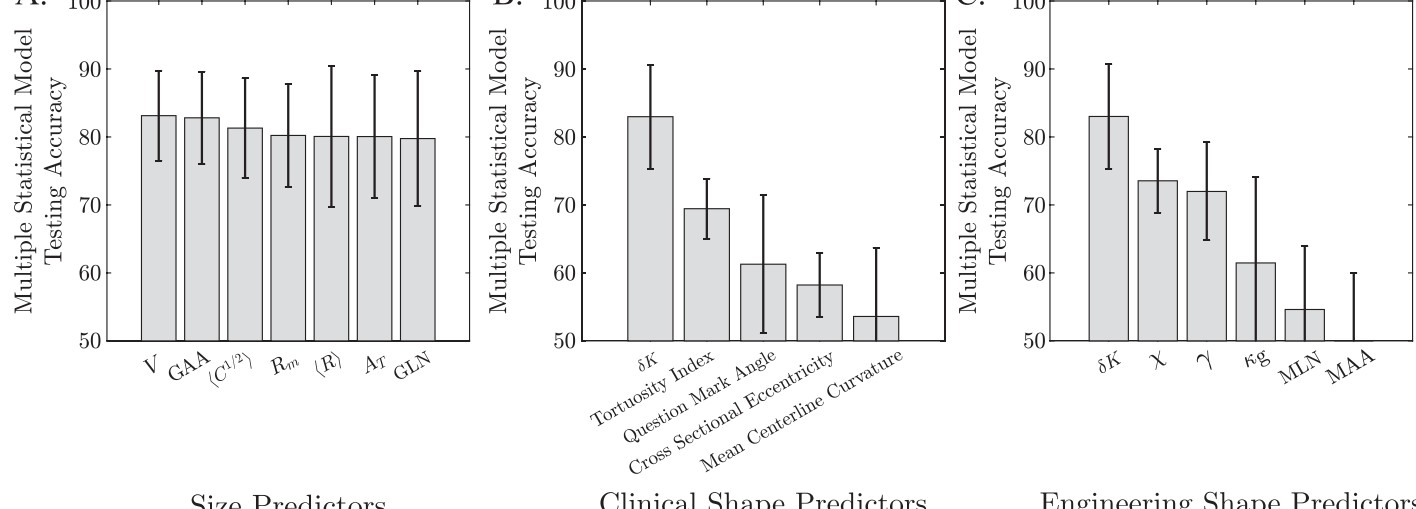

**Fig 10. Aortic Population Classification Based on Various Size and Shape Features.** Comparison of size and shape metrics in classifying aortic disease state from medical imaging. A. Measures of aortic size achieve similar classification accuracies and thus are functionally equivalent (corroborating Fig 5). The GLN and GAA are other size metrics. B. $\delta K$ significantly outperforms measures of aortic shape from the clinical literature in classifying aortic disease state (normal non-diseased aortas, diseased aortas with desired outcomes following TEVAR, and aortas with failed outcomes following TEVAR). C. $\delta K$ outperforms general shape metrics from the broad engineering literature. Error bars indicate ±1 standard deviation of the classification accuracies for the different classification methods.

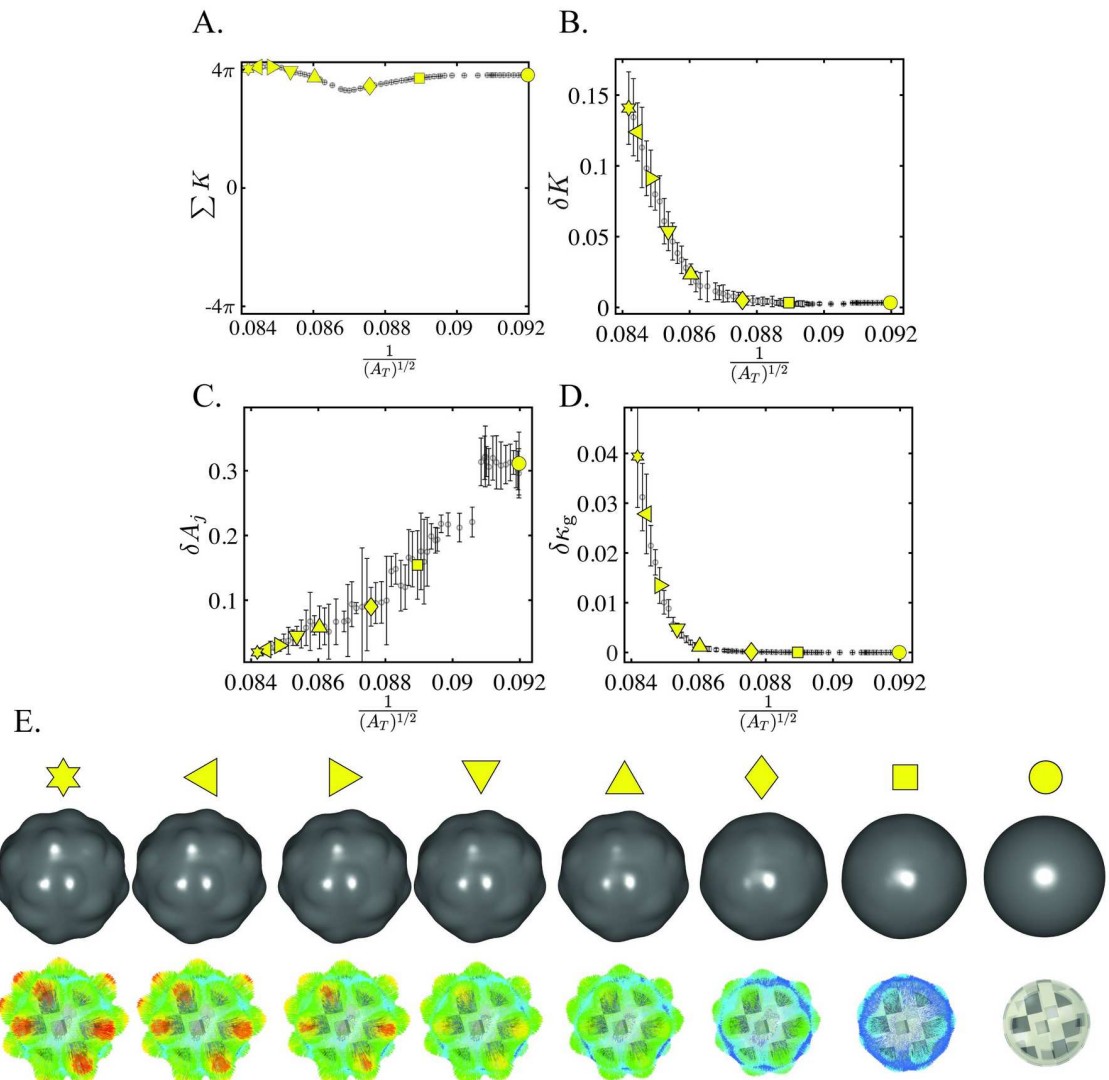

**Fig 11. $\delta K$ and $\delta \kappa_g$ Equivalent for Small Changes in Overall Size.** Simulation of a sphere with pressurization followed by growth. A. $\Sigma K$ is a topologic invariant and thus remains relatively constant throughout the simulation. B. $\delta K$ captures the increasing degeneration of the spherical surface due to growth. C. $\delta A_j$ fails to capture this degeneration. D. $\delta \kappa_g$ seems to capture the degeneration of the spherical surface as the value diverges for increasing size. However, the narrow scale of the x-axis indicates that there is little increase in overall size for this simulation. E. Surface geometries for selected frames in the simulation, with the undeformed geometry on the right side and the final geometry on the left side. The vectors indicate the direction of surface deformation.

Fig 9 compares the efficacy of a single-variable space defined by maximum diameter ($2R_m$), the clinical standard, with an enhanced shape-size feature space for predicting treatment outcomes. Using size as the sole metric of disease change, the cornerstone of imaging-based practice in other clinical contexts, is inherently problematic because a critical point between closely-spaced and overlapping populations is very sensitive to the available data. For instance, a well-known problem in $R_m$-based criteria is the bias against smaller-statured female patients because of the heavily male-weighted population-based statistics [70, 71]. This is best illustrated by Fig 9. While arbitrarily defined size classifiers did discriminate amongst normal

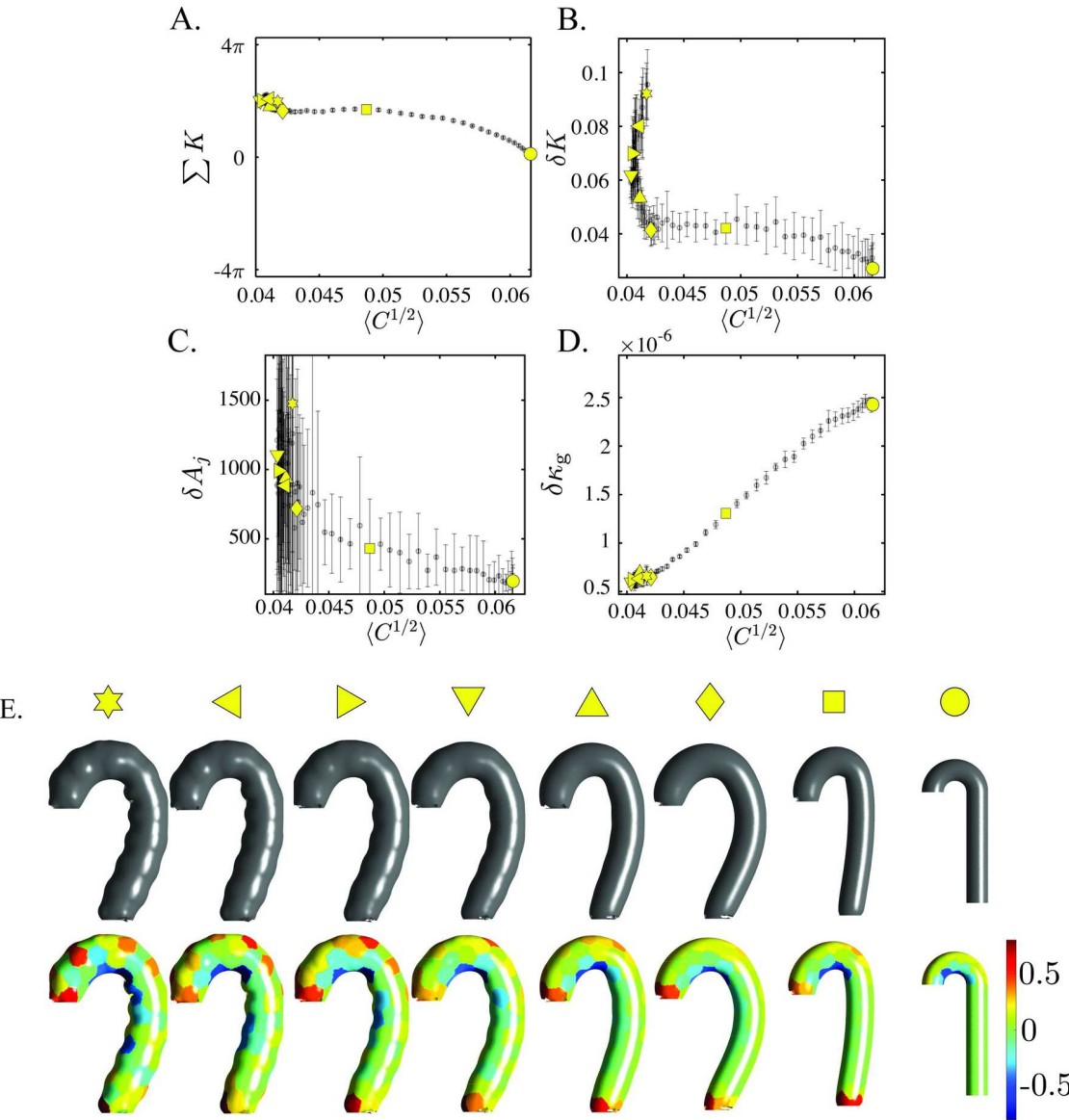

**Fig 12. $\delta K$ Superior to $\delta\kappa_g$ with Significant Size Changes.** Simulation of an ideal aorta with pressurization followed by growth. A. $\sum K$ is a topologic invariant and thus remains relatively constant as $\langle C^{1/2}\rangle$ increases throughout the simulation. B. $\delta K$ captures the increasing surface degeneration due to growth. C. $\delta A_j$ does not capture this degeneration, as evidenced by the increasing error with simulation progression. D. When size significantly changes, $\delta\kappa_g$ no longer captures the geometric deformation. E. Surface geometries for selected frames in the simulation, with the undeformed geometry on the right side and the final geometry on the left side. The heatmap coloring indicates $K_j$ the total curvature at the per-partition level.

aortas, successful TEVAR, and failed TEVAR, such scalars lack physical meaning and clinical generalizability outside of the specific cohort being analyzed. This is partly due to natural population-level variation in aortic size in addition to the inherently operator-dependent nature of aortic size measurements (such as diameter) [30].

The addition of the second axis ($\delta K$) alleviates this issue by providing a quantifiable and reproducible shape scalar. $\delta K$ also captures the global geometry of the aorta and is operator-independent. The combined $(\widetilde{\delta K}, \tilde{\ell}^{-1})$-feature space demonstrates greater than 90%

classification accuracy for the same three cohorts. The addition of a tangible shape axis also offers enhanced interpretability of the underlying geometric trends driving aortic pathology. Such a classification space can be clinically applied to pre-operative treatment planning for aortic dissection patients. $\delta K$ outperforms previously described shape metrics in both the clinical and engineering literatures. Clinically-derived shape measures, such as the tortuosity index, are predominately acquired from aortic centerlines [21, 72]. These measures underperform compared to $\delta K$ and are no better than $\ell^{-1}$ alone for characterizing aortic disease pathology from geometry. As such, future analysis of $\delta K$ is liable to demonstrate substantial clinical application of $\delta K$ as a clinical outcomes predictor.

No general theory exists on the meaning of $\delta K$ divergence. In soft matter systems such as spherical vesicles, where size weakly changes, and in dynamic systems, rapid fluctuations of Gaussian curvature have been linked to so-called 'topologic catastrophes' indicative of physical instability [48, 49, 73]. While we do not inherently study aortic stability as it relates to clinical rupture, we show that aortas with high $\delta K$ independently classify as high risk for clinical complications and poor outcomes post-TEVAR. It is therefore reasonable to conjecture that vertical divergence in the $(\widetilde{\delta K}, \tilde{\ell}^{-1})$-space is a sign of aortic instability and an indicator of suboptimal suitability for endovascular repair.

The morphologic evolution of biological structures is tightly integrated with disease development in many other contexts. $\delta K$ is a size-independent shape metric, and because it only requires extrinsic geometric information, this procedure can be applied to any surface mesh geometry. For instance, 3D imaging is extensively used to analyze lung nodules for malignancy [1], breast lesions for tumor growth [74], liver irregularities for cirrhosis [2], cerebral aneurysms [75], and the left ventricle for heart failure [76].

As with the aorta, size-based criteria form the mainstay of clinical approaches, while shape is qualitatively used but has been proven difficult to quantify until now. This methodology is based on a general geometric and topologic foundation, and future analysis will be needed to validate its extension to other clinically relevant problems of characterizing disease through analysis of shape change in medical imaging.

## Supporting information

**S1 Appendix. Supplementary information supporting the main text.** Fig A includes the demographic information for the non-pathologic aortic cohort. Fig B is the demographic information for the dissection cohort. Section titled "Aortic Segmentation and Post-Processing from CTA Imaging" includes details on the methods and procedures involved in Segmentation, Noise Reduction, Smoothing, Isolation of the Outer Surface of the aortic mask, and Meshing. The section on "Calculation of the Shape Operator" details our implementation of the Rusinkiewicz algorithm of calculating surface curvatures on a meshed surface which are the primary inputs into our shape and size calculations. The section "Artifact Removal" details the criteria used to remove the flat edges and and rims which are generated during the segmentations and constitute artifacts. The section "Jensen-Shannon Divergence of Partition Gaussian Curvature" details our implementation of the JSD as a measure of $\kappa_g$ spatial gradients within partitions. The section titled "Sensitivity to Partition Size" details our exploration of how patch size impacts the distribution of data projected into the shape-size feature space. The section "Ideal Shapes" provides the analytical functions used to generate the idea shapes used for cross-validation of our methods in the manuscript. The section "Other Shape Metrics" shows our detailed exploration of other published functions quantifying shape and the projection of our data into each one of the individual shape-size feature spaces. The section "Finite Element Simulations" provides details of material model selection and element selection for the FEA

simulations in the paper. And the final supplementary section "Analysis on Pre-Operative Data" projects only the last pre-operative scan into the shape-size feature space, this is a reduced dataset of the full data set provided in Fig 7 of the paper.
(PDF)

**S1 Dataset. Contains all the data necessary to reproduce the main figures: Scan ID, mean curvedness, $\delta K$, total aortic area, aortic volume, mean aortic radius, max aortic radius, tortuosity index, cross sectional eccentricity, mean centerline curvature, question mark angle, radius ratio, and scan sequence.** The data correspond to the raw mask segmentations which can be obtained from the public GitHub: https://github.com/SurgBioMech/khabaz_2024/tree/main.
(XLSX)

## Acknowledgments

We are grateful to Enrique Cerda, Efi Efrati, Haim Diamant, and Thomas Witten for careful reading of the text and detailed comments.

## Author Contributions

**Conceptualization:** Sanjeev Dhara, Nicole Bohr, Cheong Jun Lee, Gordon Kindlmann, Ross Milner, Luka Pocivavsek.

**Data curation:** Kameel Khabaz, Karen Yuan, Joseph Pugar, David Jiang, Sanjeev Dhara, Janet Kang, Kathleen Cao, Nicole Bohr, Cheong Jun Lee, Ross Milner, Luka Pocivavsek.

**Formal analysis:** Kameel Khabaz, Karen Yuan, Joseph Pugar, David Jiang, Seth Sankary, Sanjeev Dhara, Junsung Kim, Janet Kang, Gordon Kindlmann, Luka Pocivavsek.

**Funding acquisition:** Newell Washburn, Nicole Bohr, Cheong Jun Lee, Gordon Kindlmann, Ross Milner, Luka Pocivavsek.

**Investigation:** Kameel Khabaz, Karen Yuan, Joseph Pugar, David Jiang, Seth Sankary, Sanjeev Dhara, Junsung Kim, Gordon Kindlmann, Luka Pocivavsek.

**Methodology:** Kameel Khabaz, Nhung Nguyen, Luka Pocivavsek.

**Project administration:** Kathleen Cao, Luka Pocivavsek.

**Resources:** Ross Milner, Luka Pocivavsek.

**Software:** Kameel Khabaz, Joseph Pugar, Nhung Nguyen, Luka Pocivavsek.

**Supervision:** Joseph Pugar, Seth Sankary, Kathleen Cao, Newell Washburn, Gordon Kindlmann, Ross Milner, Luka Pocivavsek.

**Validation:** Kameel Khabaz, Joseph Pugar, Luka Pocivavsek.

**Visualization:** Kameel Khabaz, Karen Yuan, Joseph Pugar, Junsung Kim, Luka Pocivavsek.

**Writing – original draft:** Kameel Khabaz, Joseph Pugar, David Jiang, Luka Pocivavsek.

**Writing – review & editing:** Kameel Khabaz, Joseph Pugar, David Jiang, Luka Pocivavsek.

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
