## [Decision Letter · Decision Letter 0]

15 Nov 2023

Dear Professor Pocivavsek,

Thank you very much for submitting your manuscript "The Geometric Evolution of Aortic Dissections: Predicting Surgical Success using Fluctuations in Integrated Gaussian Curvature" for consideration at PLOS Computational Biology.

As with all papers reviewed by the journal, your manuscript was reviewed by members of the editorial board and by several independent reviewers. In light of the reviews (below this email), we would like to invite the resubmission of a significantly-revised version that takes into account the reviewers' comments.

The reviewers raise comments regarding details of the model and presentation of the results in the figures. Additionally, consideration of age should be added and discussed to increase impact of this work.

We cannot make any decision about publication until we have seen the revised manuscript and your response to the reviewers' comments. Your revised manuscript is also likely to be sent to reviewers for further evaluation.

Sincerely,

Alison Marsden

Academic Editor

PLOS Computational Biology

Stacey Finley

Section Editor

PLOS Computational Biology

Reviewer's Responses to Questions

**Comments to the Authors:**

Reviewer #1: In this work, methods of topology are applied to study aortic morphogenesis throughout the life-cycle of an aorta. The authors work with aortic surfaces derived from cross-sectional images from CT scans from both normal patients and patients with aortic dissections at various stages of disease (302 aortas in total). From this inspection, two clinical regimes are observed: shape preserving (normal) growth and (diseased degeneration) growth with shape changes.

This leads the authors to define a dual parameterization space of both aortic size and shape (two orthogonal metrics) by using the total curvature K as the primary measure of shape. It is first proved that all aortas scale as generalized bent cylinders parameterizable by a single length scale. It is then shown that the variance of K captures the shape evolution by characterizing local shape changes.

It is finally shown that the consideration of these combined metrics outperform other size and shape measures from the clinical and engineering literatures when applied to the predictive classification of Thoracic Endovascular Aortic Repair (TEVAR) success. It is suggested that such a classification space can be clinically applied to pre-operative treatment planning for aortic dissection patients.

The work done is original and innovative, providing relevant methods and results to researchers in the field. The manuscript could be considered for publication. The following remarks should be addressed by the authors:

P5: "We prove that all aortic shapes are homeomorphic to T^2". Please define T^2, which is mentioned several times in the manuscript.

P5: "Type IA endoleak". Do you mean 1A, which is defined below in the text? Please define here.

P6: "Isolation of the outer surface". What about the inner surface? Is the thickness variation/evolution not taken into account? Would the inner surface lead to the same conclusions?

P10. Results: Why are major/minor results (3.1)-(3.5) introduced in the order 1,2,4,3,5 and never referred to below (in that order or any other)?

P10. FE simulations: - "All parameters are determined from Fe instead of the total deformation F in the case growth is triggered." Why Fe? It is F, not Fe, what determines the observable geometry.

- Se in Eq. (10) is defined as the second Piola-Kirchhoff stress, but how it is obtained is not defined anywhere. Indeed, the actual second Piola-Kirchhoff stress is that given in Eq. (9), i.e. S.

- Why is isotropic growth with constant growth rate in the longitudinal-circumferential plane assumed? Why is the tissue not allowed to grow in the through-the-thickness direction? This is somehow related to my previous comment on P6.

Fig. 12: "Ib. dK captures the increasing surface degeneration due to growth. Ic. dAj does not capture this degeneration." It seems that dAj follows the same tendency as dK in Fig. 12. Please explain better these conclusions.

P23 FE simulations: Parameters used for the NH model give as a result a nearly incompressible behavior. The type of elements used are not specified. Please specify them and clarify if they prevent possible volumetric locking.

Reviewer #2: This review is uploaded as an attachment.

Reviewer #3: # Summary

The manuscript introduces a shape-size space for analyzing the development of the dissected aorta.

Strengths:

- Mathematically sound shape-size feature space.

- Profound analysis of various parameters and alternatives to corroborate the results presented.

- Compelling results.

Weaknesses:

- My main concern is the figures presented. When I look at Figures 1, 2, 3, 7 and 15, I have the impression that the aorta is evolving from a "normal length" aorta to a very short (longitudinally) but tortuous or dilated aorta. In other words, it seems to be getting shorter (at least visually) and ends in the descending thoracic aorta (at best). It appears to become disproportionately thicker than longer when diseased, see figure 1 on the right.

- Since the surface partitioning patch size seems to be related to the radius, the question is how this should be assessed when comparing/analyzing the development. Aren't patches of different sizes being compared/analyzed? What are the resulting implications?

The manuscript represents a substantial body of work with significant scientific innovations. I believe that my comments can be taken into account in a revision. I, therefore, advocate accepting this manuscript.

Remarks:

- eq11: I do not see the growth rate in equation 11

- figure 1: the yellow glyphs remain unclear. What is their meaning and what are they representing?

**Have the authors made all data and (if applicable) computational code underlying the findings in their manuscript fully available?**

Reviewer #1: None

Reviewer #2: **No: **The authors state that data and code is available "upon request", which does not seem to meet the PLOS Data policy stating "the data and code should be provided as part of the manuscript or supporting info, or deposited to a public repository".

Reviewer #3: **No: **as stated, available upon request

PLOS authors have the option to publish the peer review history of their article (what does this mean?). If published, this will include your full peer review and any attached files.

Reviewer #1: No

Reviewer #2: **Yes: **Hannah Cebull

Reviewer #3: No
---

## [Decision Letter · Decision Letter 1]

9 Jan 2024

Dear Professor Pocivavsek,

We are pleased to inform you that your manuscript 'The Geometric Evolution of Aortic Dissections: Predicting Surgical Success using Fluctuations in Integrated Gaussian Curvature' has been provisionally accepted for publication in PLOS Computational Biology.

Best regards,

Alison Marsden

Academic Editor

PLOS Computational Biology

Stacey Finley

Section Editor

PLOS Computational Biology

Reviewer's Responses to Questions

**Comments to the Authors:**

Reviewer #1: My comments have been addressed by the authors and the manuscript amended. To be accepted.

Reviewer #2: The authors have sufficiently addressed all of my concerns. Again, I believe this work presents a novel metric and is a good start for demonstrating the possible impact that "total curvature" may have for improving diagnoses/treatments of aortopathies.

Reviewer #3: I would like to thank the authors for their very thorough revision and cover letter. All my comments have been

sufficiently taken into account.

**Have the authors made all data and (if applicable) computational code underlying the findings in their manuscript fully available?**

Reviewer #1: None

Reviewer #2: Yes

Reviewer #3: **No: **as stated, available upon request

PLOS authors have the option to publish the peer review history of their article (what does this mean?). If published, this will include your full peer review and any attached files.

Reviewer #1: No

Reviewer #2: **Yes: **Hannah Cebull

Reviewer #3: No

---

## [Editor Report · Acceptance letter]

26 Jan 2024

PCOMPBIOL-D-23-01525R1 

The Geometric Evolution of Aortic Dissections: Predicting Surgical Success using Fluctuations in Integrated Gaussian Curvature

Dear Dr Pocivavsek,

I am pleased to inform you that your manuscript has been formally accepted for publication in PLOS Computational Biology. Your manuscript is now with our production department and you will be notified of the publication date in due course.

With kind regards,

Olena Szabo
